# Artificial Intelligence for 3D Reconstruction from 2D Panoramic X-rays to Assess Maxillary Impacted Canines

**DOI:** 10.3390/diagnostics14020196

**Published:** 2024-01-16

**Authors:** Sumeet Minhas, Tai-Hsien Wu, Do-Gyoon Kim, Si Chen, Yi-Chu Wu, Ching-Chang Ko

**Affiliations:** 1Division of Orthodontics, The Ohio State University College of Dentistry, Columbus, OH 43210, USA; 2Department of Orthodontics, Peking University School and Hospital of Stomatology, Beijing 100082, China; 3Division of Periodontology, The Ohio State University College of Dentistry, Columbus, OH 43210, USA

**Keywords:** three-dimensional reconstruction, deep learning, maxillary impacted canines, dental reconstruction

## Abstract

The objective of this study was to explore the feasibility of current 3D reconstruction in assessing the position of maxillary impacted canines from 2D panoramic X-rays. A dataset was created using pre-treatment CBCT data from a total of 123 patients, comprising 74 patients with impacted canines and 49 patients without impacted canines. From all 74 subjects, we generated a dataset containing paired 2D panoramic X-rays and pseudo-3D images. This pseudo-3D image contained information about the location of the impacted canine in the buccal/lingual, mesial/distal, and apical/coronal positions. These data were utilized to train a deep-learning reconstruction algorithm, a generative AI. The location of the crown of the maxillary impacted canine was determined based on the output of the algorithm. The reconstruction was evaluated using the structure similarity index measure (SSIM) as a metric to indicate the quality of the reconstruction. The prediction of the impacted canine’s location was assessed in both the mesiodistal and buccolingual directions. The reconstruction algorithm predicts the position of the impacted canine in the buccal, middle, or lingual position with 41% accuracy, while the mesial and distal positions are predicted with 55% accuracy. The mean SSIM for the output is 0.71, with a range of 0.63 to 0.84. Our study represents the first application of AI reconstruction output for multidisciplinary care involving orthodontists, periodontists, and maxillofacial surgeons in diagnosing and treating maxillary impacted canines. Further development of deep-learning algorithms is necessary to enhance the robustness of dental reconstruction applications.

## 1. Introduction

Maxillary canines are the second most commonly impacted teeth in the dental arch, following the third molars. The prevalence of maxillary canine impactions ranges from 1% to 3%, and of these cases, approximately 83% to 98% are unilateral impactions [1,2]. Orthodontic and surgical interventions are typically necessary, as these impacted canines can lead to the root resorption of neighboring teeth and cause functional and aesthetic problems. Successfully planning orthodontic treatment requires determining the precise position and spatial context of the impacted canine. Radiographic assessments are employed to make these diagnoses, utilizing either two-dimensional radiographs (panoramic X-ray, lateral or posterior–anterior cephalometric, and periapical X-rays) or advanced three-dimensional imaging technology like cone beam computed tomography CBCT).

Historically, 2D radiographs have been the primary tool for diagnosing the position of impacted canines. However, they suffer from limitations such as poor visibility and the potential misrepresentation of structures. Distortion, the overlay of three-dimensional structures, and imaging artifacts can adversely affect the localization and treatment planning of maxillary impacted canines [3]. Newer methods, such as CBCT, offer significant advantages. CBCT allows the orthodontist to create 3D images, providing a much more accurate localization of maxillary impacted canines from various dissection views [4]. This advanced imaging technology empowers dentists to diagnose and plan treatments with far greater accuracy and precision, enabling customized treatment plans with minimal complications. The benefits of CBCT images extend beyond implant planning, surgical assessments of pathology, temporomandibular joint (TMJ) evaluation, and pre- and post-operative assessments of craniofacial fractures [5]. However, the adoption of CBCT has been limited for several reasons. Firstly, CBCT involves higher irradiation doses compared to 2D radiographs, which has raised concerns about radiation exposure [3]. Additionally, the cost of the CBCT can be prohibitive for many patients, limiting its popularity, especially in small dental clinics or developing countries [6,7].

Over the past decades, researchers have been diligently working on advancing techniques for 3D reconstruction from 2D X-ray images [8,9]. These efforts aim to provide superior details and higher accuracy while simultaneously reducing radiation exposure and costs to patients [10,11]. In recent years, with the impressive success of deep learning in computer vision-related fields, researchers have begun exploring its potential for 3D reconstruction from 2D X-rays. For instance, Kim et al. employed a convolutional neural network (CNN) to extract informative features and combined it with statistical shape modeling (SSM) to reconstruct 3D leg bones from 2D X-rays [12]. Similarly, Kasten et al. introduced an end-to-end deep network designed to directly learn the distribution of bone shapes, bypassing the need for SSM. Their approach achieved an average Dice similarity coefficient of 0.906 when reconstructing 3D knee bones from bi-planar X-ray images [13]. 

Concurrently, Zheng et al. proposed the use of a generative adversarial network (GAN), specifically X2CT-GAN, for reconstructing a computed tomography (CT) from two orthogonal X-rays [7]. Distinct from prior studies focused on 3D model reconstruction, X2CT-GAN has the remarkable capability of generating a 3D volumetric image from just two 2D chest X-rays. While the clinical value of the reconstruction output was not reported, it holds promise for future applications where physicians may utilize this algorithm to reconstruct a CT-like 3D volume from a standard X-ray machine. In the realm of dental imaging, He et al. developed a two-step reconstruction algorithm known as Oral-3D, with the goal of reconstructing the 3D bone structure of the oral cavity using only a single panoramic X-ray [6,14]. Unlike chest X-rays, panoramic images are single-view images captured with a moving camera. Achieving 3D reconstruction from a single view necessitates additional information, such as prior knowledge about the dental arch’s shape. In the first step, a deep-learning neural network, such as a GAN, is employed to expand the 2D panoramic image to a flattened 3D representation. In the second step, this flattened image is curved along the real archform. The Oral-3D framework has demonstrated success in reconstructing healthy oral cavities, oral cavities with missing teeth, and oral cavities with wisdom teeth.

It is recognized, even in the studies performed by He et al., that AI is currently unable to produce a 3D reconstruction that is completely identical to the ground truth. To evaluate the reconstruction quality, the structural similarity index measure (SSIM) is a classical metric used in the reconstruction task. Both X2CT-GAN and Oral-3D adopted this metric and achieved approximately 72% and 78%, respectively. The current reconstruction results allow the researchers to identify wisdom and missing teeth, as demonstrated by the work of He et al. For the present study, we attempt to understand the current clinical application of AI to 3D reconstruction by solving a more complex and nuanced problem than He et al. We propose reconstructing the 3D dental structure and tooth position from 2D panoramic X-rays in patients with maxillary impacted canines. From this reconstruction output, we will assess the clinical value of the reconstruction output to determine if our algorithm can be of use to orthodontists to help diagnose and plan treatments for cases with maxillary impacted canines. 

Inspired by He et al., we also break down the task of 3D bone reconstruction via a 2D panoramic X-ray into a two-step process [14]. However, in this study, we will only focus on the first step of the reconstruction technique, as this is where the major reconstruction process occurs, and the error in this process needs to be minimized to construct an accurate 3D oral structure with bone density and the curved mandibular structure. We choose to call the result of this first step a pseudo-3D image. From this pseudo-3D image, the position of the impacted maxillary canine is located, and the clinical feasibility of our AI algorithm is determined.

The purpose of this work is to examine if an artificial intelligence algorithm achieving a certain level of computer-vision metric (e.g., SSIM) can be used for 3D reconstruction in a way that accurately preserves the clinical information needed to identify the position of a maxillary impacted canine so that an orthodontist can dependably diagnose and treatment plan for these patients.

## 2. Materials and Methods

### 2.1. Dataset

A dataset was created using pre-treatment CBCT data from a total of 123 patients, including 74 patients with impacted canines and 49 patients without impacted canines. These CBCT scans were collected from Peking University Hospital with institutional ethical committee approval (IRB: PKUSSIRB-201626016) [15]. The CBCT machine (NewTom VG, QR s.r.l., Verona, Italy) was used under the following settings: 15 × 15 cm field of view, 110 kV, and 1–20 mA (pulsed mode), with a resolution of 0.3 mm isotropic voxel and exposure time of 10 s. The exclusion criteria included (1) previous orthodontic treatment, (2) cleft palate or other maxillofacial syndromes, (3) maxillary dental/skeletal trauma or surgical history, and (4) dental age younger than the late mixed dentition. The age range of the subjects was between 11 and 18 years of age. The total numbers of males and females were 50 and 73, respectively. The average age was 14.59, with a standard deviation of 2.29. Table 1 shows the demographic summary of the subjects recruited in this study.

Corresponding 2D panoramic X-rays and pseudo-3D images were prepared for all CBCT images. Instead of using direct scans from a 2D panoramic X-ray machine, we projected these images from 3D CBCT data using 3D slicer with Sandbox module [16]. All the pseudo-3D images were also generated via 3D slicer and then resized to 128 voxels along anterior–posterior (AP) and inferior–superior (IS) directions, with a fixed spacing size of 0.3 mm in right–left (RL) direction. Therefore, the size of the 2D panoramic X-ray was N × 128, where N was the number of voxels in RL direction of the corresponding pseudo-3D image. The intensity of all voxels was normalized between −1 and 1 for both 2D and 3D images. It is recognized that synthetic panoramic images differ in intensity distribution from original scans. To address this, a representative sample was selected as a template, and we performed histogram matching on the remaining samples to align their intensity distributions more closely with the template, minimizing the differences and ensuring consistency across the dataset. This method prepared paired 2D and 3D image sets for AI training.

These data were used to train a deep-learning reconstruction algorithm (described in the Section 2.2). Of the 74 patients who were used to train the algorithm, 36 were in the buccal position, 12 in the middle, and 26 in the lingual position. In the mesial and distal direction, 65 samples were in the mesial position, and 9 samples were in the distal position. We did not define a middle position in the mesial and distal direction.

### 2.2. Deep-Learning Network

Since the absorption of radiation varies among different tissues, the depth information of the teeth and the mandible is embedded in the 2D panoramic X-ray. According to Oral-3D by He et al., the 3D reconstruction of bone structures from 2D panoramic X-ray comprised 2 stages: dimensional expansion from 2D to pseudo-3D images and deformable registration to bend the pseudo-3D image to fit the actual archform [6,14].

Regarding the first stage, we adopted X2CT-GAN framework that allows us to reconstruct a pseudo-3D image from a single 2D image [7]. This pseudo-3D image contains information related to depth, as well as the location of the impacted canine in the buccal/lingual, mesial/distal, and apical/coronal positions. In the second stage, this pseudo-3D image can be converted to a 3D structure of the oral cavity when paired with information related to the dental arch. 

For this study, we chose to focus on the first stage of reconstruction (single 2D image to a pseudo-3D image) as the major reconstruction process happens in this stage, and errors should be minimized in this stage to ensure an accurate 3D structure of the oral cavity is produced in the second stage. 

The X2CT-GAN deep generative framework was used for the 3D reconstruction of CT volume from biplanar chest X-rays [7]. The original work also proposed a variant of X2CT-GAN, which only takes a single 2D X-ray to generate the 3D chest structure. Inspired by the single X-ray variant, we proposed a deep generative framework, called Pan2CBCT, in this study to generate pseudo-3D bone structure from 2D panoramic X-ray. Similar to the standard GAN architecture, Pan2CBCT consists of two networks: generator and discriminator (Figure 1).

Inspired by Oral-3D, the loss functions used in Pan2CBCT are as follows:$$Loss_{D}^{GAN}=E_{y}{\left[D\left(y\right)-1\right)}^{2}]+E_{x}\left[D{\left(G\left(x\right)\right)}^{2}\right],\;Loss_{G}^{GAN}=E_{x}{\left(D\left(G\left(x\right)\right)-1\right)}^{2}],$$
where $Loss_{G}^{GAN}$ and $Loss_{D}^{GAN}$ represent the adversarial loss for generator and discriminator, respectively; *x* and *y* represent the panoramic and pseudo-3D images, respectively; *G* and *D* represent the generator and discriminator functions; and *E* denotes the expected value [6,14]. To further improve the generated quality, two additional loss functions in terms of voxel-wise and plane-wise regularization, $Loss_{R}$ and $Loss_{P}$, are used for generator, which are expressed as
$$Loss_{R}=E_{x,y}\left[{\left(y-G\left(x\right)\right)}^{2}\right],\;Loss_{P}=\frac{1}{3}\left[E_{x,y}\left[{\left(P_{ax}\left(y\right)-P_{ax}\left(G\left(x\right)\right)\right)}^{2}\right]+E_{x,y}\left[{\left(P_{co}\left(y\right)-P_{co}\left(G\left(x\right)\right)\right)}^{2}\right]+E_{x,y}\left[{\left(P_{sa}\left(y\right)-P_{sa}\left(G\left(x\right)\right)\right)}^{2}\right]\right],$$
where $P_{ax}$, $P_{co}$, and $P_{sa}$ are the projections in the axial, coronal, and sagittal planes, respectively. The ultimate optimization problem tries to minimize the *G* and minimize the *D* and is formulated as
$$D^{*}=arg\underset{D}{\min}Loss_{D}^{GAN},\;G^{*}=arg\underset{G}{\min}\lambda_{1}Loss_{D}^{GAN}+\lambda_{2}Loss_{R}+\lambda_{3}Loss_{P}.$$

The Pan2CBCT was trained using Adam optimizer [17]. For all impacted canine and non-impacted canine paired Pan-CBCT samples, the input to the network is a 128 × 128 2D patch of panoramic X-ray image, and the output is a 128 × 128 × 128 3D patch of reconstruction of the bony structure. In this manner of patching, a paired 2D–3D sample becomes several 3D 128 × 128 × 128 patches for training. For example, a pseudo-3D image with size of 320 × 128 × 128 with stride of 32 could contribute to 10 patches with size of 128 × 128 × 128 (Figure 2). 

The entire imaging workflow is shown in Figure 3.

The entire implementation process was conducted using Pytorch [18] on the computing resource provided by the Ohio Supercomputer Center [19]. 

The algorithm performance was evaluated using 15-fold cross-validation. The training data consisted of a total number of 123 samples: 74 samples with impacted canine and 49 samples without impacted canine. The training data with impacted canines were split into 15 groups: 14 groups with 5 samples and 1 group with 4 samples. The samples without impacted canine were kept as 1 group. Each fold contained all the training data, with 1 group serving as the test group (Figure 4).

### 2.3. Evaluation Metric and Clinical Predictability

The reconstruction was evaluated using the structure similarity index measure (SSIM) as a metric to indicate the reconstruction quality. This measure compares the two images based on 3 features: luminance (l), contrast (c), and structure (s). The SSIM is reported to be between 0 and 1. A value closer to 1 indicates that the images are very similar, whereas a value closer to 0 indicates that the images are very different. 

The accuracy of the location of the impacted canine was calculated by comparing the ground truth images and the output of the algorithm along both buccal–lingual and mesial–distal directions.

## 3. Results

The reconstruction performance was evaluated by accurately identifying the impacted canine on the output (Figure 5). In Figure 5, the impacted canine can be appreciated in both the reconstruction output and the ground truth on the buccal aspect. From visual evaluation, the differences are obvious. The sagittal, coronal, and axial slices of the reconstruction are blurry and depict a loss of detailed tooth structure compared to the ground truth. However, the anatomical structures are maintained, and the position of the impacted canine can be discerned. 

The position of the impacted canine in either the buccal, middle, or lingual position was identified with 41% accuracy (Table 2). The algorithm was trained with 36 buccal samples, 12 middle samples, and 26 lingual samples. As almost 50 percent of the training data had buccal samples, the greatest reconstruction accuracy was for the buccal samples.

In the mesial and distal positions, the reconstruction output had a 55% accuracy (Table 3). The training sample contained 65 samples in the mesial position and 9 samples in the distal position. As 88% of the training data was in the mesial position, the reconstruction outputs in the mesial direction had greater accuracy.

To quantitatively measure the similarity of the reconstruction output and the ground truth, SSIM was employed. The average SSIM value was 0.71, with a minimum of 0.63 and a maximum of 0.84. A value of 0.84 indicates a reconstruction quality that has a modest similarity to the ground truth. There was a statistical difference between the ground truth of the flattened CBCT and the AI-generated flattened CBCT based on the SSIM (*p* < 0.001). The 75%, 50%, and 25% quantiles of SSIM are 0.74, 0.71, and 0.69, respectively. The ground truth and the reconstruction output can be compared visually in Figure 6. The boundaries between the individual teeth and the bone are blurry, especially in the coronal slice compared to the ground truth. Despite having a higher SSIM value than the output of Figure 5 (SSIM = 0.68), the reconstruction output does not visually look much different. Visual image quality may be more important than the SSIM value, as the images will eventually be read visually by an orthodontist to diagnose and plan treatments for the impacted canine.

## 4. Discussion

The reconstruction of a 3D image from a 2D panoramic X-ray could be useful in various dental practices, including but not limited to orthodontic diagnosis and treatment planning of maxillary impacted canines, implant insertion, alveolar bone grafts, auto-transplantation, etc. In this study, we developed a deep-learning-based AI system to reconstruct a pseudo-3D image. From this pseudo-3D image, we were able to locate the position of the maxillary canine in either the buccal, lingual, or middle position, as well as in either the mesial or distal position on most of the samples. 

To quantify the reconstruction quality, SSIM was used as the key criterion, as it is a measure correlated to the image quality as perceived by humans. Our reconstruction output showed promising SSIM values, with an average value of 0.71 and a range of 0.63–0.84, indicating modest similarity between the reconstruction output and the ground truth image. The SSIM evaluates similarities within pixels, and if the pixels of the ground truth and the predicted reconstruction images align with similar pixel density values, the value will be closer to the positive one. 

Although our reconstruction SSIM values achieved similar values to other deep-learning algorithms in the healthcare field [6,7,14,20], we were only able to identify the position of the impacted canine in either the buccal, middle, or lingual position with 41% accuracy. These results suggest that the use of AI in 3D reconstruction for diagnosis is not yet suitable for clinical application at an SSIM level of approximately 0.7. Thorough clinical studies are crucial to ensure that the actual clinical information is correctly preserved before considering the practical application of the algorithms. Presently, reconstruction outputs with an SSIM value in the 0.7 range may be used for the identification of missing teeth or wisdom teeth but lack anatomical details for diagnosis and treatment planning. 

Our accuracy values indicate that the clinical applicability of our AI system is still lacking. Image quality degradation is observed between the ground truth and the reconstruction output. The blurry boundaries between the alveolar bone and tooth structure make it very difficult to identify the impacted canine location. Given the quality of the images, it is not feasible for an orthodontist to diagnose and plan treatments using the reconstruction output. 

In the buccal, middle, and lingual directions, the positions were identified with 41% accuracy. The greatest reconstruction accuracy was in the buccal direction, as almost 50% of the training data was in the buccal direction. This indicates that training sample size plays an important role in the accuracy of the reconstruction output. The reconstruction algorithm had trouble reconstructing samples in the lingual direction, as demonstrated by an accuracy of 12%. For samples that contained an impacted canine in the middle, the accuracy was between the buccal and lingual positions. 

The mesial and distal positions of the impacted canine were identified from the reconstruction output with 55% accuracy. The AI was able to reconstruct the impacted canine in the mesial–distal position with higher accuracy compared to the buccal–lingual direction. The AI system did not need to learn the depth information from the 2D panoramic X-ray to reconstruct the impacted canine in the mesial or distal position, leading to greater accuracy in the reconstruction output. 

Our current AI algorithm attempts to solve a much more nuanced and complicated problem that goes beyond the simple 3D reconstruction of a healthy oral cavity and abnormal dental structures (missing teeth and wisdom teeth). Our study, therefore, serves as an important step towards the development of more refined and accurate AI-driven diagnostic tools in the field of dentistry. This study, utilizing generative adversarial networks (GANs), shows promise for AI-based 3D reconstructions from 2D panoramic X-rays. Nevertheless, at an SSIM value of approximately 0.7, the results of 50% predictability of the canine position in the diagnostic image underscore the limitations of our study. The limitations include the poor image quality of panoramic X-rays converted from CBCT, the requirement of the matched CBCT-panoramic X-ray pairs, the limited sample size, and the AI algorithm itself. Nevertheless, the present study provides a feasibility test for future advancements and highlights the considerable potential of AI technologies in dental imaging. Recent newer models like diffusion-based AI may outperform GANs.

To overcome the aforementioned limitations and improve the quality of dental 3D reconstruction, we propose three research directions for the future. Firstly, additional modalities (e.g., cephalometric X-rays) can be introduced to provide more informative features in the anterior–posterior direction. Several previous studies used two or more X-ray images during reconstruction [21,22,23,24,25]. Additionally, the original work of X2CT-GAN presented an ablation study, demonstrating that a second-view X-ray significantly improved the SSIM in chest CT reconstruction. However, to reconstruct a 3D object from its 2D X-ray projections, obtaining the source position and image plane orientation in 3D space with high accuracy is crucial [8]. 

Secondly, the size and heterogeneity of the dataset need to be increased, as well as the heterogeneity of the dataset. Our results indicated that if each position of the impacted canine is not well represented, the accuracy of the reconstruction output is influenced. Finally, future research should aim to integrate more advanced AI technologies. Although generative models based on diffusion models have shown outstanding capabilities to generate high-quality images [26,27], they were less successful in preserving meaningful representation in the latent space compared to GAN, which is still an open challenge for diffusion models [28,29,30]. On the other hand, the cost of collecting and labeling data could be very expensive. Recent studies have shown that self-supervised learning is able to train a model using unlabeled samples or more effectively utilize the existing labeled data, resulting in improved model performance [31,32,33].

## 5. Conclusions

The clinical feasibility of employing a deep-learning model for 3D reconstruction from a 2D panoramic X-ray in cases of maxillary impacted canines has been demonstrated by generative images with a 70% similarity (SSIM = 0.7). However, 50% accuracy is not adequate for the practical application of this program in clinical settings. It is imperative to refine the data preparation and enhance the deep-learning algorithm to increase the precision and predictability of determining the canine’s position.

## Figures and Tables

**Figure 1 diagnostics-14-00196-f001:**
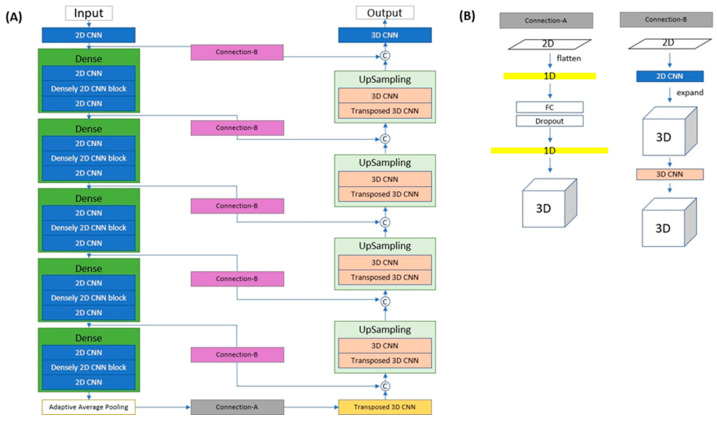
Network architecture of Pan2CBCT generator. (**A**) Encoder–decoder network and (**B**) structures of Connection-A and Connection-B.

**Figure 2 diagnostics-14-00196-f002:**
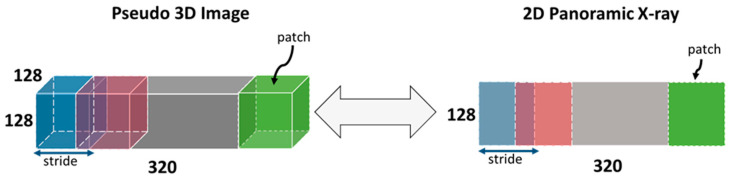
Patching of a paired 2D–3D sample.

**Figure 3 diagnostics-14-00196-f003:**
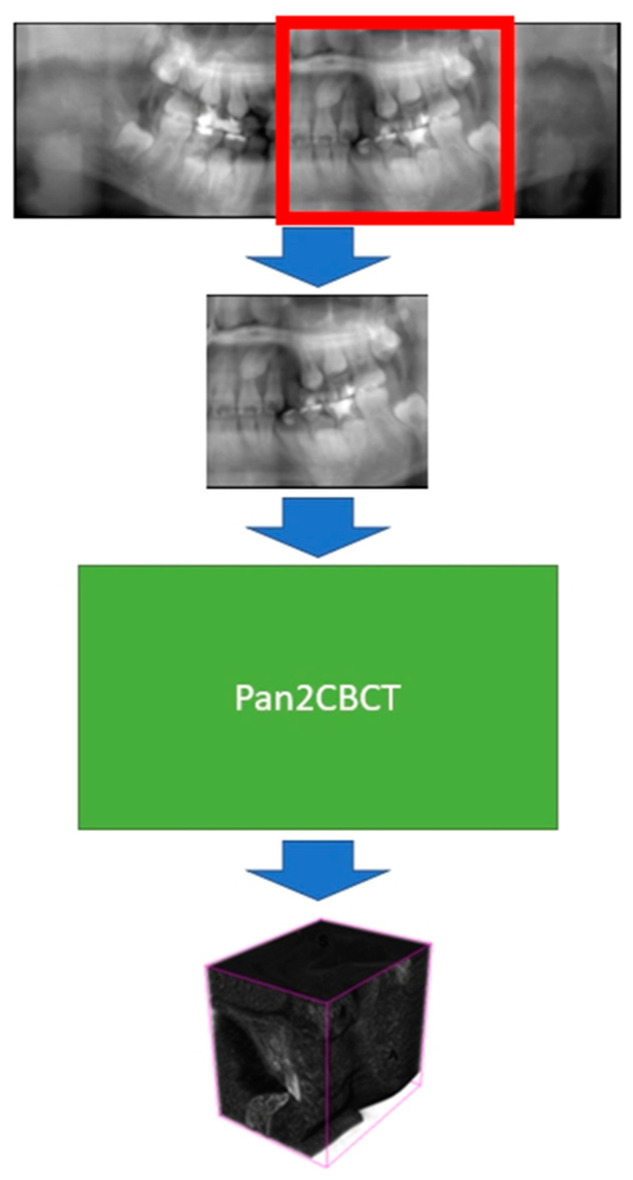
Workflow of 3D reconstruction of pseudo-3D volumetric image.

**Figure 4 diagnostics-14-00196-f004:**
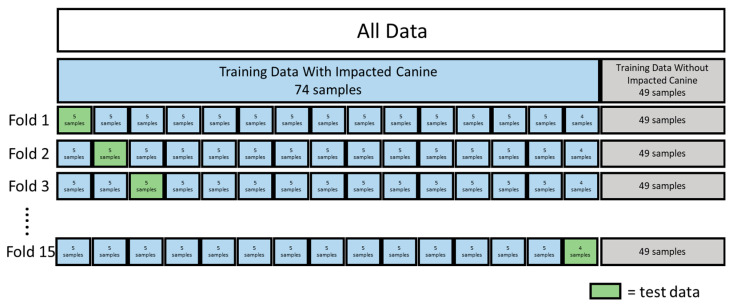
The 15-fold cross-validation for impacted canine CBCTs.

**Figure 5 diagnostics-14-00196-f005:**
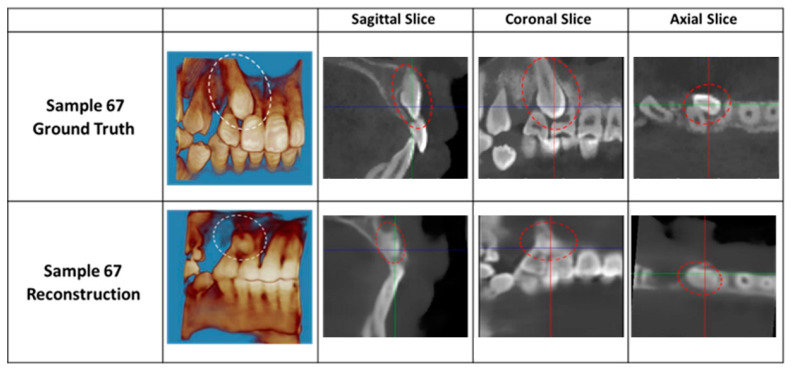
Reconstruction results compared to the ground truth. Impacted canine demarcated in either a white or red dotted circle.

**Figure 6 diagnostics-14-00196-f006:**
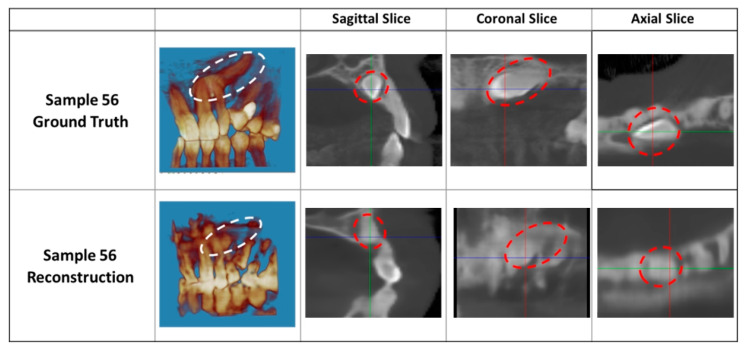
Comparison of reconstruction result with an SSIM of 0.84 and ground truth. Impacted canine demarcated in either a white or red dotted circle.

**Table 1 diagnostics-14-00196-t001:** Summary of this study’s dataset.

	Samples with Impacted Canine	Samples without Impacted Canine	Total
Number of samples	74	49	123
Mean age ± SD	14.50 ± 2.30	14.67 ± 2.29	14.59 ± 2.29
Age range	11–18	11–18	11–18
Number of Males/Females	28/46	24/25	50:73
Number of buccal/middle/lingual impactions	36/12/26	N/A	36/12/26
Number of mesial/distal impactions	65/9	N/A	65/9

SD: standard deviation; N/A: not applicable.

**Table 2 diagnostics-14-00196-t002:** Identification of buccal, middle, and lingual positions from reconstruction output compared to ground truth.

**Ground Truth Position**	**Number of Samples**	**Correct** **Identification**	**Incorrect** **Identification**	**Percentage Correct**
BUCCAL	36	23	13	64%
MIDDLE	12	4	8	33%
LINGUAL	26	3	23	12%
		Number Correct	30	
		Number Wrong	44	
		Accuracy	0.41	

**Table 3 diagnostics-14-00196-t003:** Identification of mesial and distal position from reconstruction output compared to ground truth.

**Ground Truth Position**	**Number of Samples**	**Correct** **Identification**	**Incorrect** **Identification**	**Percentage Correct**
MESIAL	65	40	25	62%
DISTAL	9	1	8	11%
		Number Correct	41	
		Number Wrong	33	
		Accuracy	0.55	

## Data Availability

Data is unavailable due to patient confidentiality.

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
