# Peer review of "Artificial Intelligence for 3D Reconstruction from 2D Panoramic X-rays to Assess Maxillary Impacted Canines"

_diagnostics, 2024, doi:10.3390/diagnostics14020196_

Round 1

Reviewer 1 Report

Comments and Suggestions for Authors

The authors propose a new method for Artificial Intelligence for 3D Reconstruction from 2D Pano- 2 ramic X-ray to Assess Maxillary Impacted Canines. After reviewing the paper, I found that the quality of the paper is up to standard. The content is well written and the results in section 3 support the conclusion in section 5. I recommend to accept the paper to be published in the journal "Diagnostics".

Author Response

Thank you so much for your positive comments.

Reviewer 2 Report

Comments and Suggestions for Authors

Thank you for submitting "Artificial Intelligence for 3D Reconstruction from 2D Panoramic X-ray to Assess Maxillary Impacted Canines"

The aim of this article was to explore the feasible of current 3D reconstruction in assessing the position of maxillary impacted canines from 2D panoramic X-rays.

The introduction is very complete and explanatory.

It is ok ethical comittee approval.

Summarize the dataset point in a table or graph to see it more clearly.

Which are the limitations of your study? Future perspective?

Do you think 50% similarity is worth anything?

When the text reference is put, they are all followed in parentheses, for example (9,10) or (1,2,5-8)

Author Response

Thank you for submitting "Artificial Intelligence for 3D Reconstruction from 2D Panoramic X-ray to Assess Maxillary Impacted Canines" The aim of this article was to explore the feasibility of current 3D reconstruction in assessing the position of maxillary impacted canines from 2D panoramic X-rays. The introduction is very complete and explanatory. It is ok ethical committee approval.

Comment #1: Summarize the dataset point in a table or graph to see it more clearly.

Response: Thank you for the great suggestion. We have added a table to clarify our dataset's presentation (Table 1) in the Method section. The key aspects such as patient demographics, number of cases with and without impacted canines, and distribution of canine positions.

Comment #2: Which are the limitations of your study? Future perspective?

Response: We acknowledge that our study has limitations, including the sample size and the scope of the AI algorithm's learning capabilities. The following descriptions of the limitations and future directions have been integrated in the last two paragraphs in the discussion section.

Limitations

This study, utilizing Generative Adversarial Networks (GANs), shows promises for AI-based 3D reconstructions from 2D panoramic X-rays. Nevertheless, the result of 50% accuracy underscore the limitations of our study, including the panoramic image quality converted from CBCT, the requirement of the matched CBCT-panoramic X-ray pairs, the limited sample size, and the algorithm itself. Recent newer models like diffusion-based AI may outperform GANs.

Future Perspective

To overcome these limitations and improve the quality of dental 3D reconstruction, we propose three directions for the future.

Firstly, additional modalities (e.g., cephalometric x-ray) can be introduced, providing more informative features such as the (AP direction) information. Several previous studies used two or more X-ray images when reconstruction [21]–[25]. Additionally, the original work of X2CT-GAN conducted an ablation study, demonstrating that a second-view X-ray significantly improved the SSIM in the chest CT reconstruction [8].

Secondly, the size and heterogeneity of the dataset needs to be increased. Our results indicated that if each position of the impacted canine is not well represented, the accuracy of the reconstruction output is influenced.

Finally, future research should aim to integrate more advanced AI technologies. Although generative models based on diffusion models have shown outstanding capabilities to generate high-quality images [26], [27], they were less successful in preserving meaningful representation in the latent space compared to GAN [28]–[30]. On the other hand, the cost of collecting and labeling data could be very expensive. Recent studies have shown that self-supervised learning is able to train a model using unlabeled samples or more effectively utilize the existing labeled data, resulting in improved model performance [31]–[33].

Comment #3. Do you think 50% similarity is worth anything?

Response: To address your comment regarding the 50% similarity, it is important to distinguish this from the Structural Similarity Index Measure (SSIM), which in our study averages at 0.71 (71%). The 50% value is specifically referred to the predictability of the canine position in the images for diagnosis. While this level of accuracy is perceived as moderate, it represents a significant stride in the application of AI for dental diagnostics. This degree of predictability provides a feasibility promise for further advancements and highlights the considerable potential of AI technologies in improving dental imaging. Our study, therefore, serves as an important step towards the development of more refined and accurate AI-driven diagnostic tools in the field of dentistry. We have revised our conclusion to reflect this implication.

Comment #4. When the text reference is put, they are all followed in parentheses, for example (9,10) or (1,2,5-8)

Response: We appreciate your keen attention to detail regarding the citation format in our manuscript. We have prepared our manuscript using the 'Diagnosis' template, which employs square brackets [] to denote in-text references. This formatting choice is in line with the template's guidelines for citation style. We assure you that all references have been meticulously checked and formatted according to these guidelines to ensure consistency and accuracy throughout the manuscript.

Reviewer 3 Report

Comments and Suggestions for Authors

Dear authors,

congratulations on the interesting topic you have chosen. Please specify the following:

-were the radiographs taken with special specifications?

-was special preparation necessary for the 3D reconstruction?

-Was there a statistically significant difference in the accuracy of the 3D reconstructions?

Author Response

Congratulations on the interesting topic you have chosen. Please specify the following:

Comment #1. Were the radiographs taken with special specifications?

Response: Thank you for your interest in our research topic. In our study, we utilized standard 3D Cone Beam Computed Tomography (CBCT) radiographs, which were acquired without any special modifications or specifications. The imaging was performed using the NewTom VG CBCT machine (QR s.r.l., Verona, Italy). The settings for these radiographs were as follows: a field of view of 15 x 15 cm, a voltage of 110 kV, and a current of 1–20 mA in pulsed mode. Additionally, the resolution was set at 0.3 mm for isotropic voxels with an exposure time of 10 seconds. These parameters were chosen to ensure optimal image quality while adhering to standard radiographic practices. This information has been added to the Method section.

Comment #2. Was special preparation necessary for the 3D reconstruction?

Response: Thanks for the opportunity to clarify our methods. We have edited the following information to the Method section.

 In this study, we employed a synthetic approach for generating 2D panoramic images. Instead of using direct scans from a 2D panoramic X-ray machine, we projected these images from 3D Cone Beam Computed Tomography (CBCT) data using the 3D Slicer Sandbox module. This method was chosen due to the challenges associated with collecting a large dataset of paired 2D and 3D images. Such synthetic strategies are increasingly common in the development of AI algorithms for dental applications, as highlighted in [14].

We recognize that synthetic panoramic images differ in intensity distribution from original scans. To address this, we selected a representative sample as a template and performed histogram matching on the remaining samples to align their intensity distributions more closely with the template, minimizing the differences and ensure consistency across the dataset.

Comments #3. Was there a statistically significant difference in the accuracy of the 3D reconstructions?

Response: Adapted from the method in the literature of imaging AI, we employed Structural Similarity Index Measure (SSIM) to evaluate the quality of our AI-based 3D reconstructions. The SSIM is used to assess image similarity between the ground truth of the flattened CBCT and the AI-generated flattened CBCT. SSIM has been used as a computer vision metric to quantify the level of similarity between the predicted images and the ground truth. Our SSIM is equal to 0.71, representing a 71% similarity, with a significant difference between the reconstructed images and the ground truth (p<0.001). The 75%, 50%, and 25% quantiles of SSIM are 0.74, 0.71, and 0.69, respectively. We have added the statement of statistical difference and quantiles ranges to the Result section. 

Round 2

Reviewer 2 Report

Comments and Suggestions for Authors

The authors have followed the changes suggested and have greatly improved the article.

Therefore, in my opinion, this scientific article meets the necessary criteria to be published in present form